# The Ultrafine-Grain Yttria-Stabilized Zirconia Reinforced β-Titanium Matrix Composites

Daria Piechowiak , Andrzej Miklaszewski * and Natalia Makuch-Dziarska

Institute of Materials Science and Engineering, Department of Materials Engineering and Technical Physics
Poznan University of Technology, Jana Pawla II 24, 61-138 Poznan, Poland;
daria.a.piechowiak@doctorate.put.poznan.pl (D.P.); natalia.makuch@put.poznan.pl (N.M.-D.)
* Correspondence: andrzej.miklaszewski@put.poznan.pl; Tel.: +48-61-665-3665

**Abstract:** Ti($\beta$) alloys have become an important class in the biomedical field due to low Young's modulus, excellent physical properties, and biocompatibility. However, their properties, like biocompatibility and, also, low wear resistance, can be still enhanced. To improve those properties, a composites approach can be applied. This research shows a new approach of the composite structure fabrication by powder metallurgy methods which for a stabile yttria-stabilized zirconia (YSZ) reinforcement phase could be obtained in the ultra-fine grain range beta-titanium matrix. In this work, the composites based on ultrafine-grain Ti-$x$Mo ($x$ = 23 wt%, 27 wt%, 35 wt%) alloys with addition 3 wt%, 5 wt% or 10 wt% YSZ, and 1 wt% $Y_2O_3$ were fabricated by the mechanical alloying and hot-pressing approach. Obtained composites were characterized in terms of their phase composition, microstructure, Young's modulus, hardness, surface free energy (SFE), and corrosion resistance. The structure of composites consists of phases based on Ti–Mo, Ti($\alpha$), and YSZ. The oxide (YSZ) powder tends to agglomerate during processing, which is revealed in composites based on Ti23Mo and Ti27Mo. However, composites based on Ti35Mo are characterized by a high degree of dispersibility and this influences significantly the hardness value of the composites obtained. Only in the case of composites based on Ti35Mo, the decrease in Young' Modulus is observed. All composites possess a hydrophilic surface property and good corrosion resistance.

**Keywords:** metal matrix composites; ultrafine-grain structure; beta Ti; YSZ; mechanical alloying





## 1. Introduction

Metals, like stainless steel, Co-based alloys, or Ti-based alloys, are used in the biomedical field. However, titanium and titanium alloys are the most commonly used materials in biomedical applications [1,2]. Their properties, such as low Young's modulus, high corrosion resistance, and outstanding biocompatibility, make these materials those most often selected for hard tissue implants [3,4]. Until now, the most widely used had been Ti-6Al-4V, but progressive researches indicate that vanadium is considered cytotoxic [3]. In excess vanadium ions in the human body can cause neurodegenerative dysfunctions like Alzheimer's disease. Moreover, they have a negative influence on fibroblast viability. Furthermore, aluminium is also hazardous to human health, it can cause bone and nerve cell damage, which, in consequence, can lead to brain and blood vessel damage. Moreover, by having the high Young modulus and mismatching with bone modulus, Ti-6Al-4V can cause a stress shielding effect [5].

Current research is focused on enhancing the biocompatibility of titanium alloys, removing toxic elements, and decreasing Young's modulus [1,6–8]. The aforementioned requirements are met by Ti($\beta$) or near Ti($\beta$) alloys [9]. These biomaterials form an important class in the biomedical field due to their containing nontoxic elements and, also, have low elastic modulus, high strength, good corrosion resistance, and excellent biocompatibility from among biometals [10,11]. Commonly used as beta stabilizing elements are Nb, Ta, Mo [12]. Molybdenum is the most effective additive of the abovemen-

tioned [10,13]. To improve the properties of Ti alloys neutral elements such as Zr or Sn are also used [10,14]. Examples of β titanium alloys are Ti-5Mo-5Ag [15], Ti-$x$Mo ($x$-23–35%) [16,17], Ti13Nb13Zr [18], Ti20Nb13Zr [19], Ti-6Al-2Nb-2Ta-1Mo [20], Ti-15Mo-Zr [13], HEA-TiNbTaMoZr [20].

It is without doubt that the surgical application of Ti–Mo has contributed to it being well-studied among researchers. The detailed study done by the current researchers proved that the properties of Ti–Mo alloys can be adjusted according to the Mo content meaning that with the increase of the Mo concentration, the rise of the hardness and the Young's modulus decrease can be observed [21]. Such an observation provides the possibility to adjust properties to the application. The stabilization of the Ti(β) phase can be enabled by adding approximately 10% of Mo in terms of a high cooling rate [16]. It is worth mentioning that there is not only a dependency between the properties of alloys and the chemical composition but also between the aforementioned alloys and the microstructure modification [6]. The properties can be improved by nanostructurization, which can be received by the mechanical alloying (MA) process. With insights from the recent studies, it can be concluded that the nanostructurization of titanium considerably improves the mechanical properties and biocompatibility [22]. The above research presents new prospects in this field, which can lead to an improvement in the better mechanical and physicochemical properties.

The disadvantage of titanium alloys is their low wear resistance [4]. This can be improved by creating composites based on titanium alloys. The reinforcing phase in the form of hard particles increases the hardness and abrasion resistance of the material [23–25].

On the other hand, compared with bioceramic and biopolymer materials, bio metallic materials possess low biocompatibility and osseointegration. To improve these properties, the following approaches could be applied: a surface modification or a volumetric composite approach [26,27].

In the composites approach, the materials mainly used are hydroxyapatite [28], bioglass [29,30], zirconium oxides ($ZrO_2$), or silicon oxides ($SiO_2$) [26]. Composites based on pure titanium and $ZrO_2$ proved enhanced strength and biocompatibility [26].

Zirconium oxide ($ZrO_2$) is a commonly used bioceramic in a dental application. $ZrO_2$ possesses high strength parameters, good corrosion resistance. Moreover, the aforementioned bioceramic does not present toxic, mutagenic, and carcinogenic effects [31]. Zirconium oxides occur in three allotropic forms. Monoclinic structure stabile to 1170 °C. At this temperature allotropic transformation appears and zirconium oxide transforms to a tetragonal structure, while above 2370 °C it transforms to a cubic structure. In terms of medical application, the tetragonal structure of zirconium oxide is used, because of its chemical stability [32,33].

Tetragonal structure in room temperature is obtained by adding stabilizing elements like magnesium oxide [34], calcium oxide [34], cerium oxide [35,36] or the most common yttrium oxide [37]. Yttria-stabilized zirconia (YSZ) is received by adding 2–9% of $Y_2O_3$ to $ZrO_2$ during the stabilization process to obtain a solid solution [38–40].

This work aimed at the fabrication of the ultrafine-grain composites based on the titanium beta matrix with YSZ + $Y_2O_3$ reinforcement and their characterization. Three alloys with different content of molybdenum were used as a matrix: Ti23Mo, Ti27Mo, Ti35Mo, and its influence on the technological aspects of the composite structure fabrication was evaluated. The variable amount of YSZ (3%, 5%, and 10%) in the composite structure was investigated and its influence on phase structural stability. The final properties such as Young's modulus, hardness, surface free energy (SFE), and corrosion resistance were also revealed in this work. Structural characterization of the obtained materials allows presenting the relation in the phase transformation dependency between the matrix type and YSZ amount, pointing also to processing technological problems like particle size range and even distribution of the reinforcement phase. During the composite fabrication, other obstacles were revealed as the decomposition of the reinforcement oxide phase and the diffusion of its components into the titanium lattice. This work proposes an additional

step that relies on the 1 wt% yttrium oxide presence and allows limiting the decomposition of zirconium oxide during hot pressing.

Referring to the present state of knowledge [41,42] and efforts [43,44] made until now, this research shows a new approach of the composite structure fabrication by powder metallurgy methods, which for a stabile YSZ reinforcement phase could be obtained in the ultra-fine grain range beta-titanium matrix. This work evaluated an approach to composite structure fabrication with the additional properties characterization that shows a promising perspective for biomedical materials. The obtained results point out, also, that further detailed research needs to prove the value of the proposed concept, especially in the range of the wear resistance and cell contact with their activity in reference to commercially available solutions.

## 2. Materials and Methods

### 2.1. Sample Preparation

The powders of Ti–Mo alloys were prepared by mechanical alloying. The primary powders were used titanium (44 μm, 99.9% purity, CAS:7440-32-6, Aesar, Karlsruhe, Germany) and molybdenum (44 μm, 99.6%, CAS:7430-08-7 Sigma Aldrich, Karlsruhe, Germany). Mechanical alloying was realized using a SPEX 800 Mixer Mill (SPEX SamplePrep, Metuchen, NJ, USA). Process parameters were as follows:

- Milling time-48 h
- Ball-to-powder ratio (BPR)-10:1
- Atmosphere-argon.

Preparation of powder composition, loading, and unloading to stainless steel vials were executed in the glove box with an argon atmosphere (Labmaster 130). The procedure was carried out on Ti23Mo, Ti27Mo, and Ti35Mo.

In the next step, the powder of the mechanically alloyed matrix and commercially available reinforcement YSZ and $Y_2O_3$ addition powder were mixed using an agate mortar until the system was homogeneous. As a reinforcement YSZ (0.1–2 μm, GoodFellow, Huntingdon, England) and $Y_2O_3$ (<50 nm, CAS: 1314-36-9, Sigma Aldrich, Karlsruhe, Germany) were used. The bulk composites were processed by powder metallurgy by the application of the hot processing approach. Hot pressing was carried out at 800 °C for 300 s with an acting pressure of 3 kN in a vacuum condition (<50 Pa). The composition prepared by this procedure is shown in Table 1.

**Table 1.** Bulk samples indication.

| Sample | Symbol |
|---|---|
| Ti23Mo | 23 |
| Ti23Mo + 3% YSZ + 1% $Y_2O_3$ | 23–3 |
| Ti23Mo + 5% YSZ + 1% $Y_2O_3$ | 23–5 |
| Ti23Mo + 10% YSZ + 1% $Y_2O_3$ | 23–10 |
| Ti27Mo | 27 |
| Ti27Mo + 3% YSZ + 1% $Y_2O_3$ | 27–3 |
| Ti27Mo + 5% YSZ + 1% $Y_2O_3$ | 27–5 |
| Ti27Mo + 10% YSZ + 1% $Y_2O_3$ | 27–10 |
| Ti35Mo | 35 |
| Ti35Mo + 3% YSZ + 1% $Y_2O_3$ | 35–3 |
| Ti35Mo + 5% YSZ + 1% $Y_2O_3$ | 35–5 |
| Ti35Mo + 10% YSZ + 1% $Y_2O_3$ | 35–10 |

### 2.2. Materials Characterization

The crystallographic structure of the sample during different processing stages was considered using X-ray diffraction (XRD, Panalytical Empyrean, Almelo, Netherlands) equipment with the copper anode (CuKα—1.54 Å) at a Brag-Brentano reflection mode

configuration with 45 kV and 40 mA parameters. The measurement parameters were set up for 20–90° with a 15 s. per step 0.0167° in all cases.

The following structural models were used for characterization:

(1) Mechanically alloyed powders:

- $Ti_{0.9}Mo_{0.1}$—ref. code 04-018-6034
- $Ti_{0.33}Mo_{0.67}$—ref. code 04-002-9769
- $Ti_{0.67}Mo_{0.33}$—ref. code 04-017-8941

(2) Sintered specimens

- $Ti(\alpha)$—ref. code 04-008-4973
- $Zr_{0.72}Y_{0.28}O_{1.862}$—ref. code 01-077-2114
- $Ti_{0.9}Mo_{0.1}$—ref. code 04-018-6034

   ➢ Ti23Mo

- Ti0.67Mo0.33—ref. code 04-017-8941
- $Ti(\beta)$—ref. code 04-003-7297
- Ti0.857Mo0.143—ref. code 01-086-2611

   ➢ Ti27Mo

- $Ti_{0.67}Mo_{0.33}$—ref. code 04-017-8941
- $Ti(\beta)$—ref. code 04-003-7297
- $Ti_{0.857}Mo_{0.143}$—ref. code 01-086-2611
- $Y_2O_3$—ref. code 04-002-2584

   ➢ Ti35Mo

- $Ti_{0.6}Mo_{0.4}$—ref. code 04-020-8692
- $Ti_{0.93}Zr_{0.07}$—ref. code 04-019-4073

Before the microstructure analysis, the samples were polished and etched with the Kroll reagent. A scanning electron microscope (SEM, MIRA3, Tescan, Brno, Czech Republic) were used to characterize the microstructure. The observation was carried out using various magnifications. For an X-ray microanalysis, an Energy Dispersive Spectrometer Ultimax 65 detector (Oxford Instruments, Abingdon, Oxfordshire, UK) was used.

### 2.3. Nanoindentation Test and Microhardness

The composites were characterized in terms of indentation hardness ($HV_i$) and Young's modulus (EIT) using the Anton Paar nanohardness NHT2 tester, with a Berkovich tip. The measurements were based on the Oliver and Pharr approach and DIN 50 359/ISO 14.577 standard. Parameters of the test were as follows: max load 200.00 mN, loading rate 400.00 mN/min, unloading rate 400.00 mN/min, pause 5.0 s.

The commercially available rod titanium (Grade 2) was used as a reference sample.

The Vickers microhardness ($HV_{0.3}$) measurements were conducted using the microhardness tester (INNOVOTEST Nexus 4302, Maastricht, Netherlands) based on ISO 6507-1 standard. For each polished sample, 10 measurements were carried out along the sample cross-section.

Measurement parameters for microhardness testing were:

- Load: 300 g,
- load operating time: 10 s.

### 2.4. Wetting and Free Surfaces Energy Analysis

The surface energy measurement was carried out on the Drop Shape Analyzer—DSA25 instrumentation (KRÜSS Scientific-Kruss, Hamburg, Germany) and the KRÜSS ADVANCE software (version 1.5.1.0, KRÜSS Scientific-Kruss, Hamburg, Germany). The surface free energy was determined by measuring the contact angle (CA) according to the Owens, Wendt, Rabel, and Kaelble (OWRK) model. KRÜSS ADVANCE 1.5.1.0 software was used to determine the surface free energy (SFE). The measurements were conducted

on polished samples. Distilled water and glycerol, whose values of surface free energies are presented in Table 2, were used as measuring liquids.

**Table 2.** The surface tension parameters of measuring liquids.

| Liquids | $\gamma L$ (mN/m) | $\gamma Ld$ (mN/m) | $\gamma Lp$ (mN/m) |
|---|---|---|---|
| Distilled water | 72.8 | 21.8 | 51.0 |
| Glycerol | 62.7 | 21.2 | 41.5 |

The parameters used were as follows: drop volume 1.5 μL, dosing speed 0.2 mL/min, measuring time 5 s, probing frequency for multiple measurements 10 fps, base cut off-automatic/manual, CA fitting method Young Laplace, measurement conditions ambient 21.5 °C.

### 2.5. Corrosion Resistance Analysis

Corrosion tests were carried out on the surfaces of all samples. Before starting the tests, the tested surfaces were ground on sandpaper with a gradation from 600 to 1200, then polished by $Al_2O_3$ suspension and cleaned to remove any possible contamination.

Measurements were carried out using the potentiodynamic approach method in a Ringer's solution (NaCl: 9 g/L, KCl: 0.42 g/L, $CaC_{l2}$: 0.48 g/L, $NaHCO_3$: 0.2 g/L) using the SOLARTRON 1285 potentiostat (Solartron Analytical, Farnborough, UK).

Test parameters for the corrosion resistance analysis were: sample area 0.5 cm$^2$, reference electrode-graphite, counter electrode-platinum, range of the tested potential 2–3 V, scanning speed 1 mV/s.

The test sample was placed in the electrolyte. Before starting the potentiodynamic tests, the open-circuit voltage (OCP) was measured. After the OCP voltage was stabilized, the tested voltage was corrected by the value of the OCP voltage.

At the end of the test, the samples were cleaned in a stream of distilled water and dried. The obtained results were developed using the CorrView software and Tafel extrapolation method for the corrosion current and potential estimation.

## 3. Results and Discussion

In the present study, the ultrafine-grain composites based on Ti-$x$Mo ($x$ = 23, 27, 35) matrix with YSZ and $Y_2O_3$ reinforcement were fabricated by mechanical alloying (MA) and powder metallurgy (PM) methods. In the next step, specimens were examined in terms of phase transformation, microstructure, mechanical properties, surface-free energy, and corrosion behaviour.

### 3.1. Structural and Morphological Powder Analysis

Figure 1 shows the XRD patterns after 48 h MA of Ti23Mo, Ti27Mo, and Ti37Mo. XRD analysis of Ti23Mo and Ti27Mo powders shows the presence of $Ti_{0.9}Mo_{0.1}$ and $Ti_{0.33}Mo_{0.67}$ phases, however, both phases possess cubic structure. Ti35Mo powder is a single-phase and is defined as $Ti_{0.67}Mo_{0.33}$ phase with cubic structure.

Figure 2 presents the scanning electron microscope (SEM) microphotographs of Ti-$x$Mo powders after 48 h of MA. In all cases, the morphology of powders particles is irregular. Furthermore, the chemical composition of powders has a considerable influence on the particle size after the mechanical alloying process. An increase in molybdenum content from 23% to 27%, and then to 35% results in significant fragmentation of the powder, where the average powder size is 116.8 ± 15.9 μm, 32.3 ± 6.9 μm, 9.3 ± 2.7 μm respectively. The abovementioned relation could also be observed in other research where the growing amount of element or the second phase in MA processing increases the speed of the synthesis that manifests in higher lattice density [16,17].

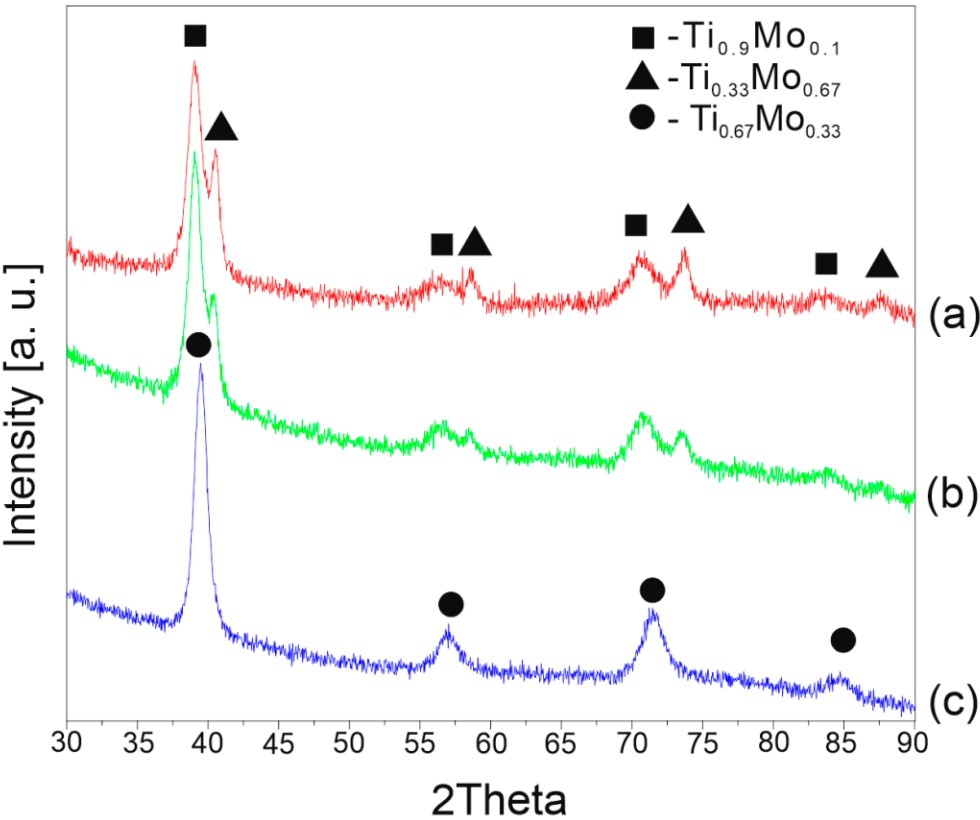

**Figure 1.** X-ray diffraction (XRD) patterns of Ti23Mo (**a**), Ti27Mo (**b**), Ti35Mo (**c**) after 48 h of mechanical alloying (MA).

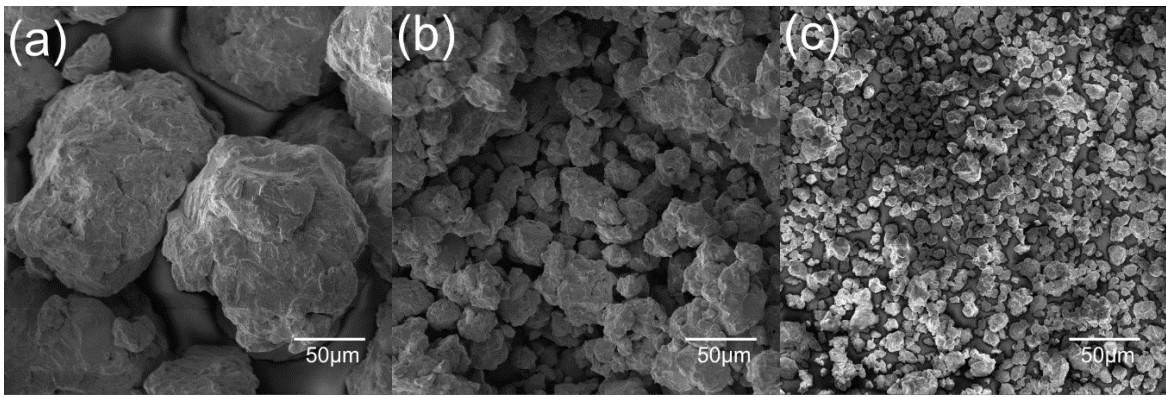

**Figure 2.** Scanning electron microscope (SEM) microphotographs of Ti23Mo (**a**), Ti27Mo (**b**), Ti35Mo (**c**) powders after 48 h of MA.

Figure 3 presents the XRD patterns of 35–10 mixed composite powder before sintering and its starting components. The mixed powder consists of $Ti_{0.67}Mo_{0.33}$, YSZ, and $Y_2O_3$ phases. The SEM microphotographs of mixed composites powder are presented in Figure 4. The particles of the ceramic phase evenly coat the particles of the metallic phase.

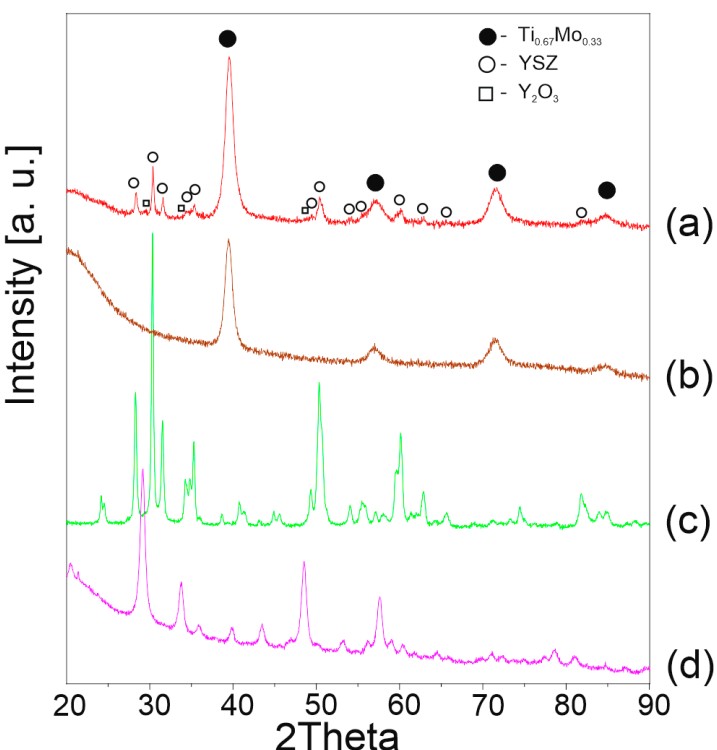

**Figure 3.** XRD patterns of 35–10 composite powder after mixing with YSZ—before sintering (**a**), and after MA (**b**) with additionally revealed YSZ (**c**) and $Y_2O_3$ powder (**d**).

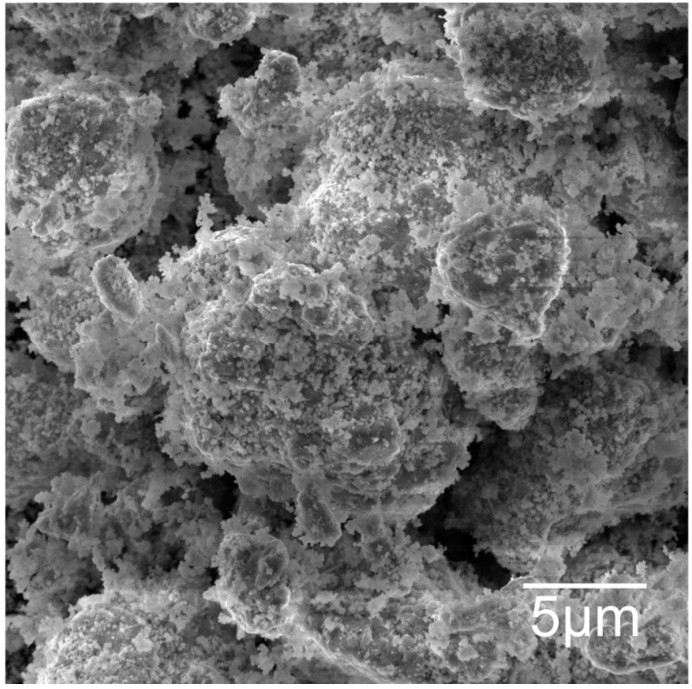

**Figure 4.** SEM microphotographs of Ti35Mo + 10% yttria-stabilized zirconia (YSZ) + 1% $Y_2O_3$ powder after mixing.

### 3.2. Structural and Microstructural Analysis of Bulk Composites

Figures 5–7 present the XRD patterns of sintered composites and base alloys as references. In all cases of composites, except the phases derived from the titanium matrix, the YSZ phase is detected. Obtained patterns confirm that the growing molybdenum

content in the matrix corresponds to a shift in diffraction angle–lower cell dimension which, however, arises as a transition to a molybdenum base cubic structure observed also in our earlier research [8] and detailed investigated structurally [17]. Additionally collected data show that the growing reinforcement share enhanced the abovementioned behaviour by the transitional phase creation also as an increase in the amount of the Ti($\alpha$) phase related to its peak intensity growth. The alpha phase content increases with the increase in the zirconium oxide content result from the oxide partially decomposition during sintering. As a consequence of this decomposition, oxygen atoms diffuse into the matrix lattice and the alpha phase is stabilized. Other relations, however, remain also visible from the patterns. A growing amount of molybdenum in the matrix constricts (partly by the size range of the starting powders) the YSZ dissolution, and the most probable explanation could be the reaction kinetics that for highly saturated solid solution show lower activity with interstitials and in the same lower intensity of the Ti($\alpha$) than for he Ti-35Mo sample emerging in $Ti_{0.93}Zr_{0.07}$ phase appearance. The abovementioned statement could also be confirmed by the yttrium oxide phase appearance for 27–10 sample, which for the disused constriction corresponds only partially, however, prove that the phase is not dissolved but separated despite ongoing kinetics.

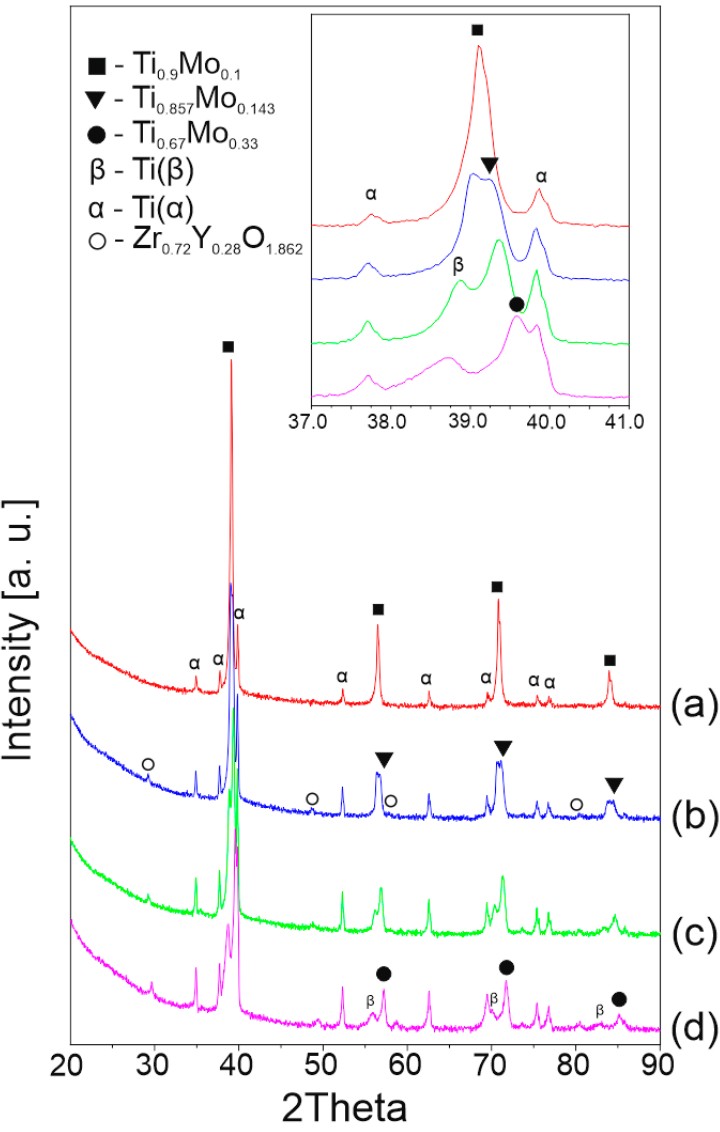

**Figure 5.** XRD patterns of 23 (**a**), 23–3 (**b**), 23–5 (**c**), 23–10 (**d**) after sintering.

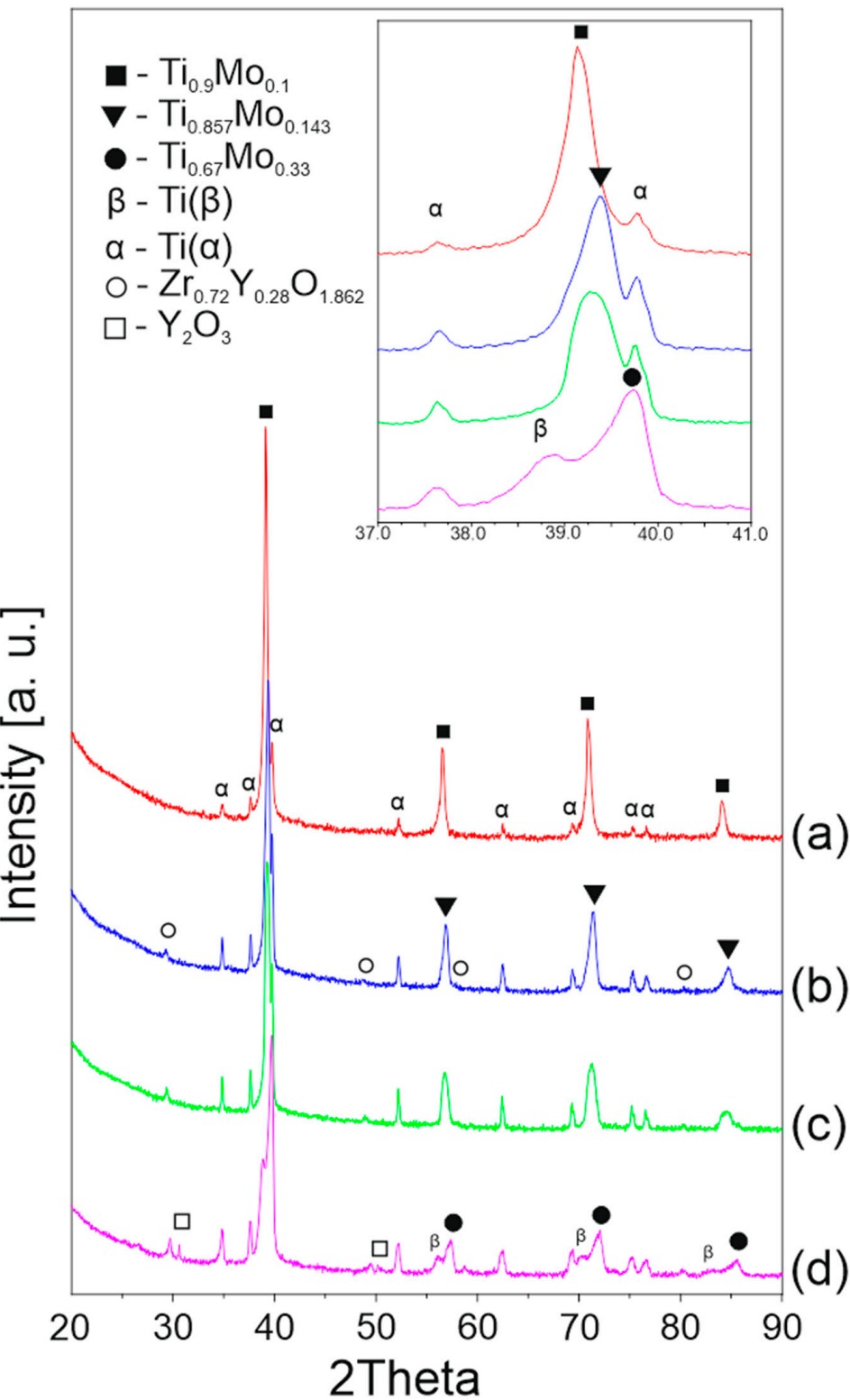

**Figure 6.** XRD patterns of 27 (**a**), 27–3 (**b**), 27–5 (**c**), 27–10 (**d**) after sintering.

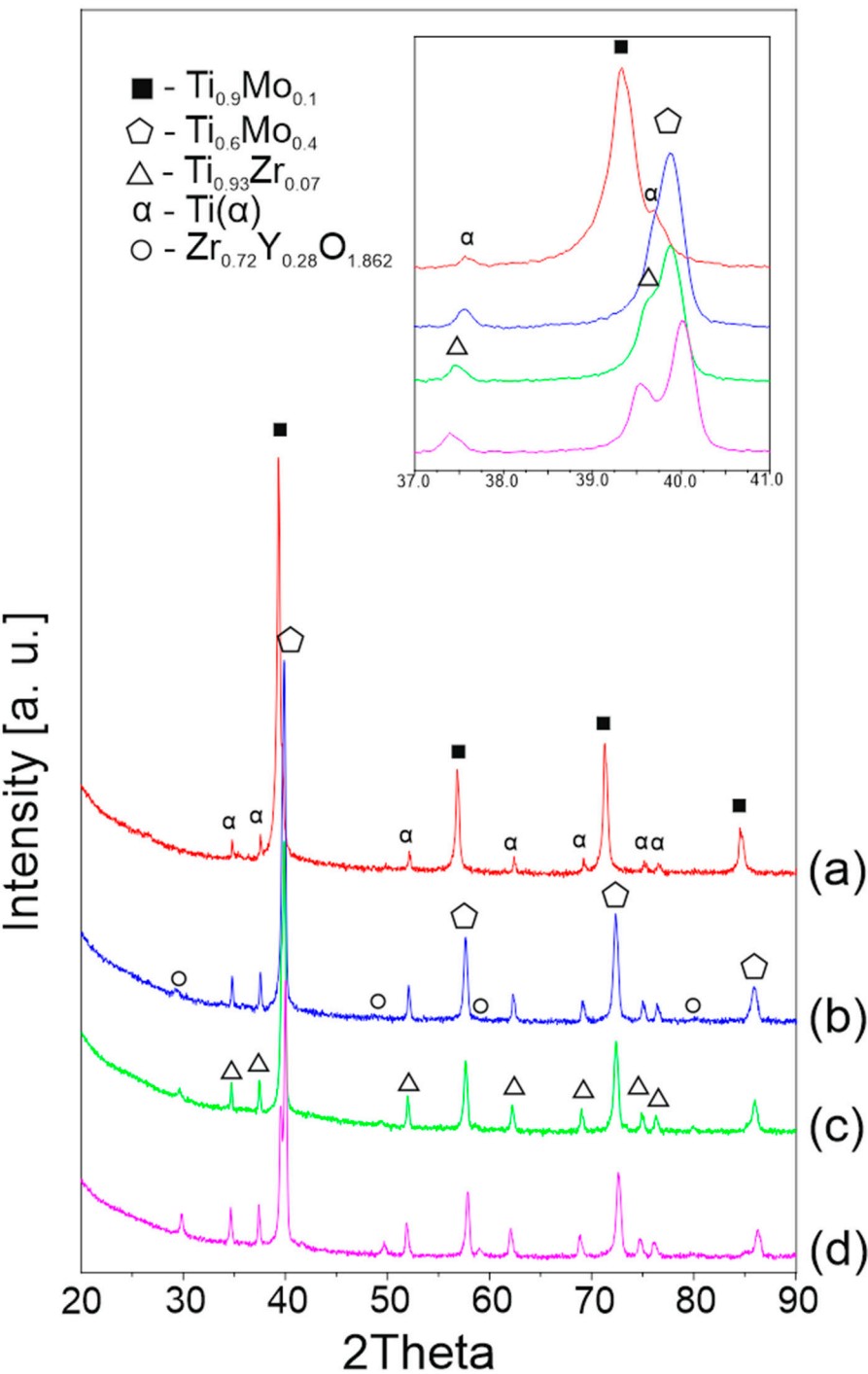

**Figure 7.** XRD patterns of 35 (**a**), 35–3 (**b**), 35–5 (**c**), 35–10 (**d**) after sintering.

Figure 8 presents the SEM microphotographs of sintered composites. Polygonal grains of the Ti (β) phase could be observed in the matrix for higher magnification in Figure 9. The reinforcement of composites is formed into aggregated YSZ grains and is characterized by a spherical morphology. Furthermore, aggregates are distributed in the areas of the grain boundaries of the mechanically alloyed particles of the matrix. The size of the powder after synthesis has a significant influence on the dispersion of the reinforcement phase particles. The most effective dispersion and the smallest aggregates of YSZ were obtained for composites based on the Ti35Mo matrix as Figure 8 confirms. The average size of aggregates determined based on Figure 9 amounts to $700 \pm 120$ nm.

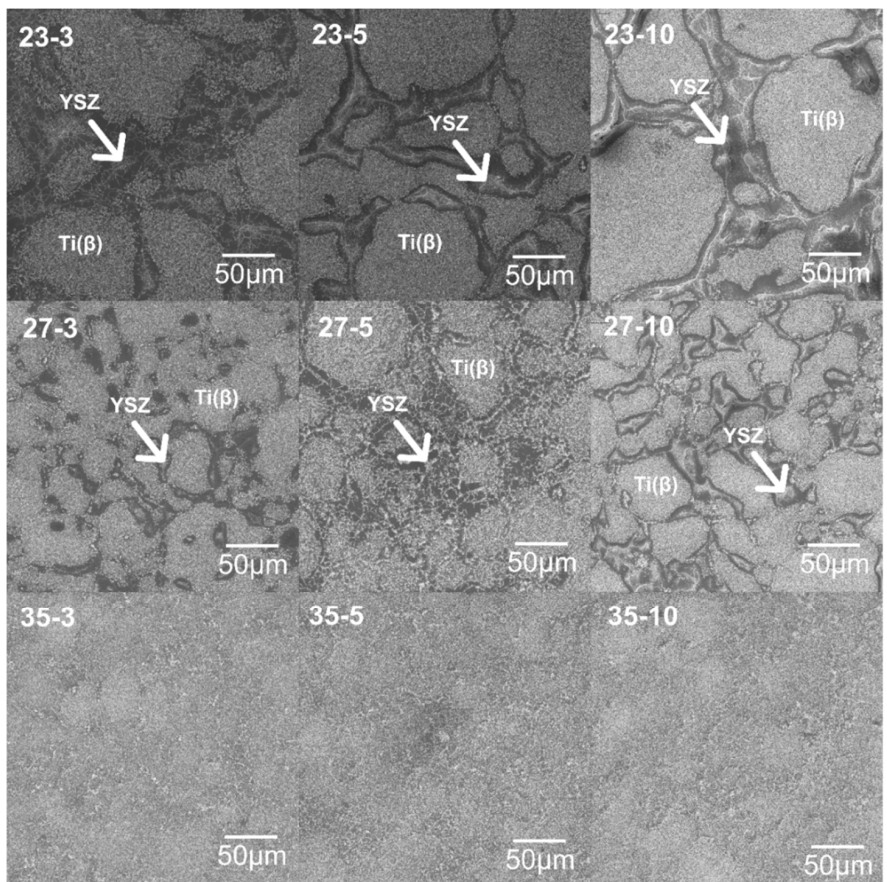

**Figure 8.** SEM microphotographs of sintered composites with visible darker initial powder grains boundary regions of YSZ aggregation.

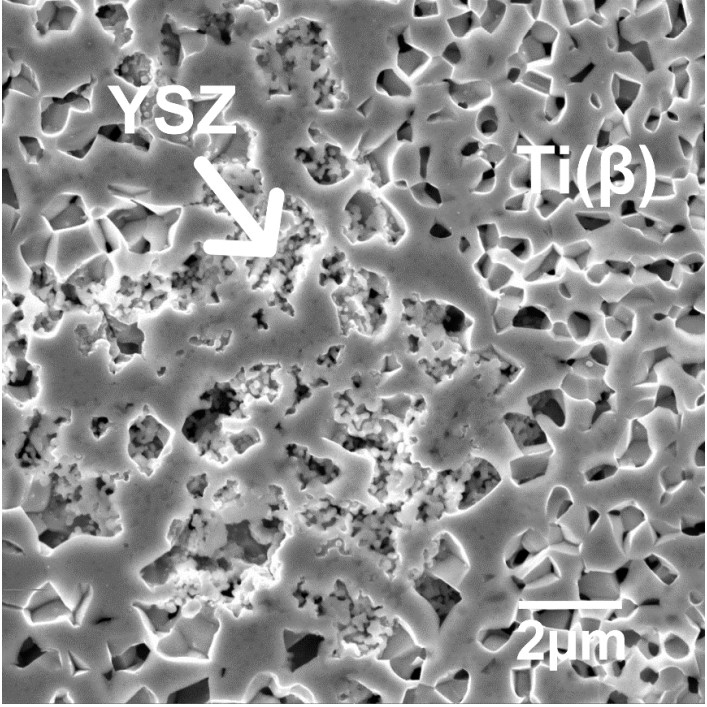

**Figure 9.** SEM microphotograph of the 35–10 specimen.

The results obtained show that the zirconium oxide partially decomposes during sintering as a result of the treatment. The zirconium atoms and oxygen diffuse into the titanium crystal lattice [42]. The zirconium remains neutral for the lattice structure, but the oxygen stabilizes the alpha phase [43]. Moreover, the presence of oxide phases in the structure contributes to the division of the $Ti_{0.1}Mo_{0.9}$ phase into $Ti(\beta)$, $Ti_{0.857}Mo_{0.143}$, $Ti_{0.67}Mo_{0.33}$ for composites based on Ti23Mo and Ti27Mo. In the case of Ti35Mo composites, the $Ti_{0.1}Mo_{0.9}$ phase is transformed into $Ti_{0.6}Mo_{0.4}$. The reason for this dependence can be found in the local diffusion of molybdenum atoms from the areas of stabilized $Ti(\alpha)$ (Figure 10). The presence of Fe atoms confirms the MA process impurity, however, Fe is also a $Ti(\beta)$ stabilizing element and can be found in β-type alloys [45]. In this case, Fe is dissolved in Ti–Mo matrix and is not revealed on XRD spectra. In the case of other matrices, the iron content remained at the same level. Further research is needed to confirm the Fe influence on the process and obtained properties; however, in the first place efforts should consider the elimination of the impurity by WC reactor and cryomilling trial.

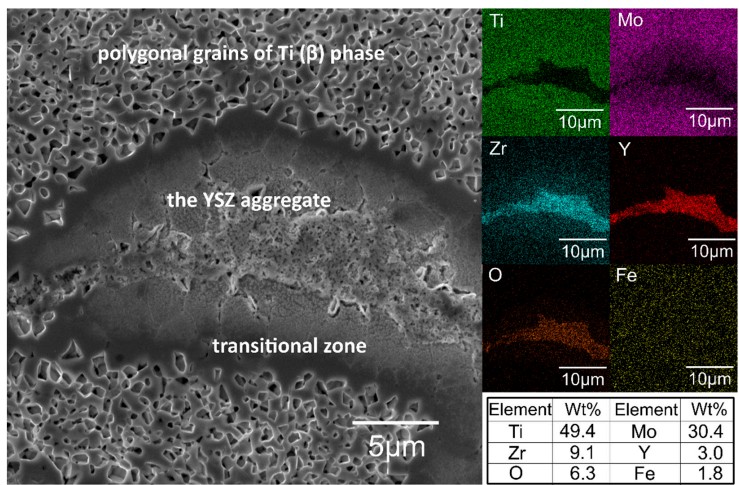

**Figure 10.** SEM microphotograph and EDS analysis of the 27–10 specimen.

After sintering, the yttrium oxide phase is visible only in 27–10 XRD pattern. The reason for the lack of the yttria phase in other patterns is that yttrium oxide diffuses in the matrix crystal lattice or act as a stabilizer of YSZ. For the purposely selected 27–10 sample, for EDS mapping it could be observed also the third possibility, confirmed earlier structurally. Yttria oxide shows a transition point that has not been dissolved in the matrix lattice and YSZ, that is reliant, however, on the molybdenum saturation, microstructure size, and YSZ addition.

### 3.3. Nanoindentation Test and Microhardness

Young's modulus and hardness of Ti–Mo composites and, also, of pure titanium, as a reference, are presented in Table 3. The indentation modulus of bases alloys reached average values of 126.5 GPa, 134.4 GPa, and 148.5 GPa respectively for Ti23Mo, Ti27Mo, and Ti35Mo. A similar trend was observed in references [17]. The general tendency is as follows with an increase in the content of the reinforcement phase, Young's modulus increases, however for samples with the highest analyzed reinforcement addition results deviate from the average most for Ti23Mo and Ti27Mo samples. On the other hand, the most uniform results connected with the microstructural homogeneity and size were obtained for the composites based on Ti35Mo which for the general trend cannot be seen. Composites 27–10 has the highest value of Young's modulus, which equals 163.2 GPa. Furthermore, the highest rise in the value of modules can be observed for composites based on the Ti27Mo matrix, and the abovementioned remains related to the phase composition, its size, and the mutual arrangement in the composite microstructure obtained. The above factors influence the measurements and estimated values also a nanohardness, based on the same Oliver

and Pharr approach that represents close to modulus trend. This limitation and lack of general judgment force the Vickers microhardness test for evaluation mean that averaged results provide a better general overview.

**Table 3.** Young's modulus and hardness of Ti–Mo composites and pure titanium.

| Sample | EIT $\pm\ \sigma$ (GPa) | HV $\pm\ \sigma$ | HV$_{0.3}\ \pm\ \sigma$ |
|--------|------------------------|------------------|-------------------------|
| Ti | 123.3 $\pm$ 14.3 | 170 $\pm$ 19 | 174 $\pm$ 13 |
| 23 | 126.5 $\pm$ 3.2 | 503 $\pm$ 11 | 447 $\pm$ 8 |
| 23–3 | 136.3 $\pm$ 9.2 | 497 $\pm$ 59 | 545 $\pm$ 44 |
| 23–5 | 140.7 $\pm$ 6.9 | 542 $\pm$ 87 | 552 $\pm$ 21 |
| 23–10 | 144.3 $\pm$ 14.6 | 525 $\pm$ 114 | 678 $\pm$ 54 |
| 27 | 134.4 $\pm$ 4.9 | 332 $\pm$ 26 | 463 $\pm$ 18 |
| 27–3 | 146.4 $\pm$ 2.5 | 530 $\pm$ 39 | 541 $\pm$ 20 |
| 27–5 | 143.6 $\pm$ 12.0 | 588 $\pm$ 40 | 619 $\pm$ 22 |
| 27–10 | 163.2 $\pm$ 25.6 | 537 $\pm$ 176 | 683 $\pm$ 57 |
| 35 | 148.5 $\pm$ 7.8 | 446 $\pm$ 39 | 507 $\pm$ 11 |
| 35–3 | 147.2 $\pm$ 8.7 | 578 $\pm$ 25 | 622 $\pm$ 10 |
| 35–5 | 149.4 $\pm$ 4.8 | 612 $\pm$ 39 | 645 $\pm$ 18 |
| 35–10 | 143.0 $\pm$ 4.9 | 522 $\pm$ 41 | 705 $\pm$ 25 |

The microhardness of specimens is enhanced with YSZ addition. Moreover, grain refinement and structural homogeneity also contribute to an increase in mechanical properties. The specimens based on Ti35Mo are characterized by the highest hardness, which reaches the value of 507 HV$_{0.3}$ for Ti35Mo and rises to 705 HV$_{0.3}$ for 35–10. In comparison, the hardness of microcrystalline pure titanium equals 174 HV$_{0.3}$.

*3.4. Surface Wettability Analysis*

The wettability and SFE of an implant's surface are important issues in the biomedical field [46]. They determine the interaction between the surface of the biomaterial and the biological environment. The hydrophilicity feature of the surface enhances the adsorption of osteoblasts and osteointegration. The contact angle of 60° is considered the cut-off point between a hydrophilic and a hydrophobic surface [45,46].

Table 4 presents the contact angles in distilled water and glycerol as well as the surface free energy for all fabricated specimens. The contact angle for all samples is lower than 60° for both water and glycerol. It indicates that all specimens possess hydrophilic surface properties. The results obtained for each Ti-*x*Mo set are similar. Surface free energies fluctuate between 47.16 mN/m for 27–10 and 70.67 mN/m for 27.

**Table 4.** Water and glycerol contact angle (CA), surface free energy with a dispersed, and polar component.

| Sample | Water CA (°) | Glycerol CA (°) | Surface Free Energy (mN/m) | Disperse (mN/m) | Polar (mN/m) |
|--------|-------------|-----------------|---------------------------|-----------------|--------------|
| 23 | 36.59 (±0.16) | 50.51 (±3.74) | 68.11 $\pm$ 8.20 | 2.91 $\pm$ 2.00 | 65.20 $\pm$ 6.20 |
| 23–3 | 33.87 (±0.11) | 42.62 (±3.09) | 64.95 $\pm$ 6.46 | 6.78 $\pm$ 2.22 | 58.17 $\pm$ 4.25 |
| 23–5 | 32.01 (±0.79) | 44.02 (±2.85) | 68.30 $\pm$ 6.23 | 5.02 $\pm$ 1.83 | 63.28 $\pm$ 4.39 |
| 23–10 | 39.11 (±0.20) | 45.78 (±2.97) | 60.47 $\pm$ 6.46 | 7.18 $\pm$ 2.32 | 53.29 $\pm$ 4.14 |
| 27 | 34.19 (±0.19) | 49.48 (±4.26) | 70.67 $\pm$ 9.26 | 2.66 $\pm$ 2.15 | 68.01 $\pm$ 7.11 |
| 27–3 | 43.64 (±1.26) | 48.87 (±5.07) | 56.50 $\pm$ 11.81 | 7.40 $\pm$ 4.30 | 49.09 $\pm$ 7.51 |
| 27–5 | 43.31 (±0.36) | 44.44 (±4.28 | 55.01 $\pm$ 9.39 | 11.14 $\pm$ 4.08 | 43.87 $\pm$ 5.32 |
| 27–10 | 52.45 (±1.13) | 48.52 (±3.94) | 47.16 $\pm$ 9.68 | 14.99 $\pm$ 4.79 | 32.17 $\pm$ 4.89 |
| 35 | 39.74 (±0.13) | 49.10 (±2.85) | 61.99 $\pm$ 6.30 | 5.06 $\pm$ 1.97 | 56.93 $\pm$ 4.33 |
| 35–3 | 47.22 (±0.32) | 39.65 (±2.89) | 51.73 $\pm$ 6.17 | 19.84 $\pm$ 3.36 | 31.89 $\pm$ 2.82 |
| 35–5 | 36.24 (±2.85) | 40.46 (±3.17) | 61.10 $\pm$ 9.35 | 9.76 $\pm$ 3.15 | 51.34 $\pm$ 6.1 |
| 35–10 | 50.06 (±0.15) | 45.74 (±3.04) | 49.10 $\pm$ 6.87 | 15.79 $\pm$ 3.52 | 33.31 $\pm$ 3.35 |

### 3.5. Surface Corrosion Resistance Analysis

The potentiodynamic polarization curves of all composites investigated in the Ringer solution are shown in Figure 11 and, also, the corrosion potential ($E_{corr}$) and corrosion current ($I_{corr}$) are presented in Table 5. The corrosion current of Ti23Mo, Ti27Mo, and Ti35Mo is as follows 1.2978 ($10^{-7}$ A·cm$^{-2}$), 8.7883 ($10^{-7}$ A·cm$^{-2}$), 8.0063 ($10^{-7}$ A·cm$^{-2}$). In contrast, the corrosion current of titanium is 1.6 ($10^{-7}$ A·cm$^{-2}$) [47].

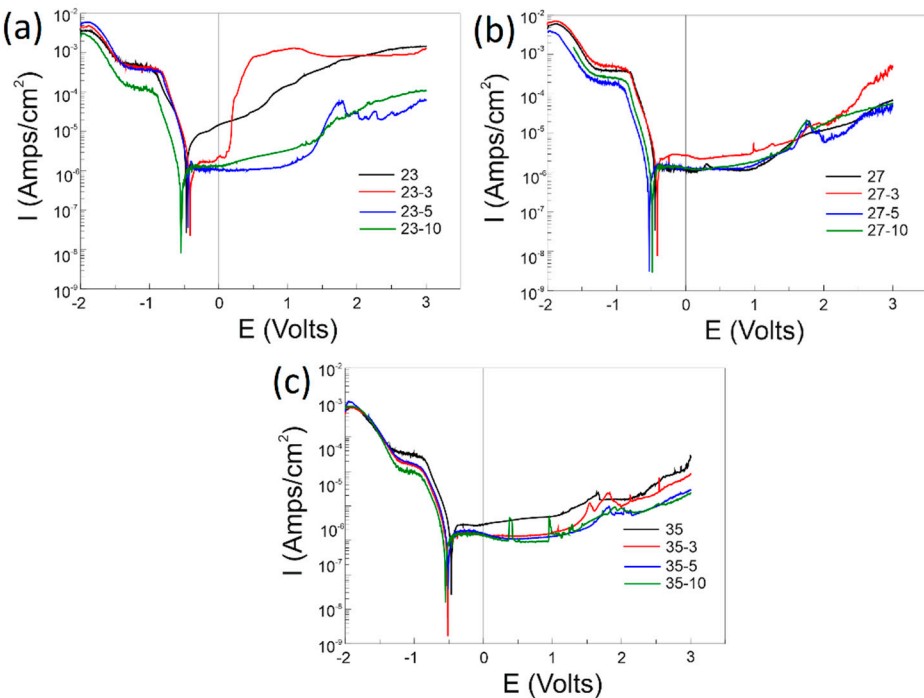

**Figure 11.** Potentiodynamic polarization curves of composites based on Ti23Mo (**a**), Ti27Mo (**b**), and Ti35Mo (**c**).

**Table 5.** Corrosion potential ($E_{corr}$) and current density ($I_{corr}$) of all sintered specimens.

| Sample | $E_{corr}$ (V) | $I_{corr}$ ($10^{-7}$ A·cm$^{-2}$) |
|---|---|---|
| 23 | −0.47156 | 1.2978 |
| 23–3 | −0.41636 | 4.0679 |
| 23–5 | −0.44637 | 5.9423 |
| 23–10 | −0.54237 | 2.9754 |
| 27 | −0.43916 | 8.7883 |
| 27–3 | −0.40917 | 6.0742 |
| 27–5 | −0.52555 | 6.0629 |
| 27–10 | −0.48249 | 5.4389 |
| 35 | −0.46641 | 8.0063 |
| 35–3 | −0.51548 | 4.6376 |
| 35–5 | −0.50704 | 9.0438 |
| 35–10 | −0.4155 | 4.7374 |

The addition of YSZ in the Ti23Mo example causes an increase in corrosion current, however, a potential change could also be observed. In the case of the Ti27Mo, the addition of YSZ results in a decline of corrosion current and for 27–10 equals approximately 5.44 ($10^{-7}$ A·cm$^{-2}$). Furthermore, the addition of 3% and 10% of YSZ to Ti35Mo reduces corrosion current. Comparatively, 5% YSZ causes an increase in corrosion current. In conclusion, the lowest corrosion current was achieved for Ti23Mo out of all specimens. In the case of composites, the best corrosion current characteristic was achieved for 23-10 specimens.

## 4. Conclusions

This work aimed at the fabrication of ultrafine-grain biocomposites based on Ti-$x$Mo ($x$ = 23 wt%, 27 wt%, and 35 wt%) with various additions of YSZ and, also, 1 wt% $Y_2O_3$. The matrix alloys were extensively analyzed in previous work of our team [16,17]. In the next step, the composites obtained were investigated in terms of the effect of the additive on phase composition, microstructure, mechanical properties, surface wettability, and corrosion resistance. The following conclusions were drawn from the conducted research:

- the increase in the content of the YSZ reinforcement contributes to the stabilization of the Ti($\alpha$) phase for the fabrication path analyzed in this work;
- the YSZ reinforcement aggregates and distribute in the areas of the grain boundaries of the mechanically alloyed particle of the matrix;
- the size of the powder after synthesis has a significant influence on the dispersion of the reinforcement phase in the sintered stage and further properties characteristics;
- in the boundary areas to the oxide phase, Mo diffuses towards the matrix and contribute to the formation of cubic phases with different contents of Ti and Mo;
- the addition of 1 wt% $Y_2O_3$ contributes to improving the stability of the zirconium oxide during sintering;
- the microstructure of the obtained composites shows an ultrafine-grain range;
- the starting powder size decreases with the increase of Mo content which yields the best dispersion for composites based on Ti35Mo;
- in the case of Ti35Mo composites, both the matrix and the YSZ additive remain in the ultra-fine grained range;
- the YSZ addition causes Young's modulus and hardness to increase;
- all of the specimens possess a hydrophilic surface characteristic;
- in the cases of Ti27Mo and Ti35Mo sets (except 35-5), the addition of an oxide reduces the corrosion current in comparison to base Ti-$x$Mo ($x$ = 27 wt% and 35 wt%) alloys

**Author Contributions:** Conceptualization, D.P. and A.M.; Data curation, D.P. and A.M.; Formal analysis, A.M.; Funding acquisition, A.M.; Investigation, D.P., N.M.-D. and A.M.; Methodology, A.M.; Project administration, A.M.; Supervision, A.M.; Visualization, D.P.; Writing—original draft, D.P. and A.M.; Writing—review & editing, A.M. All authors have read and agreed to the published version of the manuscript.

**Funding:** The work has been financed by the National Science Centre Poland under decision no.: DEC-2017/25/B/ST8/02494.

**Institutional Review Board Statement:** Not applicable.

**Informed Consent Statement:** Not applicable.

**Data Availability Statement:** Not applicable.

**Acknowledgments:** A.M. acknowledges the financial support of the National Science Centre Poland under decision no.: DEC-2017/25/B/ST8/02494. D.P. acknowledge the financial support of the Polish Ministry of Science and Higher Education (Project No. 0513/SBAD/4606).

**Conflicts of Interest:** The authors declare no conflict of interest. The funders had no role in the design of the study; in the collection, analyses, or interpretation of data; in the writing of the manuscript, or in the decision to publish the results.

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
