# Peer review of "The Ultrafine-Grain Yttria-Stabilized Zirconia Reinforced β-Titanium Matrix Composites"

_metals, doi:10.3390/met11020240_

Round 1

Reviewer 1 Report

The manuscript describes an interesting set of results on YSZ reinforced beta-titanium matrix composites. However, some aspects should be improved before publication, namely:
1. In some ways, the work looks more like a technical report than a scientific paper. Some results do not follow a general trend and the reasons that explain this behavior are not well discussed. This lack of discussion is frequent in the manuscript and should be reviewed. For example, the decrease in nanohardness observed for composites with 10% YSZ should be addressed. Another example is when the authors refer (lines 274-275) that Young's modulus increases with the amount of reinforcement, but this is not observed for composites with 35% YSZ (see table 3), and no attempt is made to explain this discrepancy.
2. The description of the powders used is incomplete. The powder granulometry is not indicated for some of them.
3. The last conclusion states that "in the cases of Ti27Mo and Ti35Mo sets except 35-5, the addition of an oxide reduces the corrosion current compared to pure Ti-Mo alloys". However, no results concerning the corrosion current density of Ti-Mo alloys are presented.
4. In lines 290 to 294 are indicated characteristics that implant surfaces must present without any bibliographic reference.
5. In lines 298-299 is stated, “Surface free energies fluctuate between 47.16 mN/m for 27-10 and 68.30 mN/m for 23-5”. However, the 27 sample has a value of 70.67 mN/m (table 4).
6. In the manuscript, figure 10 is not all visible.
7. References should be reviewed carefully. Some do not have the Journal in which they were published (ex: [3] and [11]), others have the name of the article badly (ex: [4] and [16]), others are incomplete (ex: [9]), and the volume and pages are missing in some (ex: [14] and [22]).
8. Finally, the English language must be revised because some sentences are poorly constructed.

Author Response

Dear Reviewer 1,

We would like to appreciate you allowing us to resubmit our manuscript: The ultrafine-grain YSZ reinforced beta-titanium matrix composites to Metals. We are pleased with your comments and objective feedback considering the draft of our article. The changes were marked in our article in red colors.

All of the suggestions were included during the correction of our article and the comments were appropriately responded:

In some ways, the work looks more like a technical report than a scientific paper. Some results do not follow a general trend and the reasons that explain this behavior are not well discussed. This lack of discussion is frequent in the manuscript and should be reviewed. For example, the decrease in nanohardness observed for composites with 10% YSZ should be addressed. Another example is when the authors refer (lines 274-275) that Young's modulus increases with the amount of reinforcement, but this is not observed for composites with 35% YSZ (see table 3), and no attempt is made to explain this discrepancy.

Thank you for this comment, proper changes that explain address issues were made in the manuscript text .(see p14 .l. 301-318)

The description of the powders used is incomplete. The powder granulometry is not indicated for some of them.

Thank you for this comment, correction was made in the manuscript text (see p. 3. line109-122)

The last conclusion states that "in the cases of Ti27Mo and Ti35Mo sets except 35-5, the addition of an oxide reduces the corrosion current compared to pure Ti-Mo alloys". However, no results concerning the corrosion current density of Ti-Mo alloys are presented.

Thank you for this comment the sentence was changed.

In lines 290 to 294 are indicated characteristics that implant surfaces must present without any bibliographic reference.

We appreciate that comment. We have added references to text.

In lines 298-299 is stated, “Surface free energies fluctuate between 47.16 mN/m for 27-10 and 68.30 mN/m for 23-5”. However, the 27 sample has a value of 70.67 mN/m (table 4).

Thank you for your suggestion. We have corrected that mistake.

In the manuscript, figure 10 is not all visible.

We agree with the provided comment. We enlarge the figure in the manuscript text.  

References should be reviewed carefully. Some do not have the Journal in which they were published (ex: [3] and [11]), others have the name of the article badly (ex: [4] and [16]), others are incomplete (ex: [9]), and the volume and pages are missing in some (ex: [14] and [22]).

Thank you for this comment the reference was reviewed carefully this time, only for explanation previously the Mendeley plug was used for reference insertion.

Finally, the English language must be revised because some sentences are poorly constructed.

Thank you for this comment the final text was revised by the native speaker for the improvements.

We would like to thank you once again for your suggestions for improving our manuscript.

Yours faithfully,

  1. Miklaszewski

Reviewer 2 Report

Dear Authors,

Please receive my comments concerning the presented manuscript:

[1] - p.2, line84: 2-9% of Y2O3 instead of 2-9% of Y2O5

[2] - p.2, line 86: For being in conformity with Table 1, the phrase “based on titanium beta matrix with YSZ reinforcement” should be corrected in “based on titanium beta matrix with [YSZ and Y2O3] reinforcement” because, beside YSZ with variable content, it has been used also 1% of Y2O3 as reinforcement phase. That is understandable from Table 1. If no, please revise.

[3] - for the presented experiments, the introduction of a variable amount of YSZ in the composite approach is explained at the end of the introduction. However, it is not explained the additional presence of 1% of Y2O3, beside the YSZ. What was the reason?

[4] - p.2, lines 63-64: “The disadvantage of titanium alloys is their low wear resistance [4]. It can be improved by creating composites based on titanium alloys.” It follows from this sentence that the wear resistance represents the mechanical property considered for the present research for which composite structures are created. Therefore, it must be explained why corrosion resistance and wettability were analyzed instead of wear resistance. The declaration from the final sentence of the introduction part “The influence of variable among of YSZ (3%, 5% and 10%) in composites on properties was investigated” is too general. What properties and for what reason? What was the goal for choosing specific properties to analyze? Please provide a detail explanation.

[5] - in my opinion, the introduction part fails to present the specific results regarding the already use of YSZ in composite structures. What are the actual achieving and what can be done next?

[6] - The caption from figure 3 is not quite understandable. Please revise the expression in English.

[7] - p.5, lines 201-204: For the powders Ti23Mo and Ti27Mo, the authors indicate the cubic structure for the Ti0.9Mo0.1 and Ti0.33Mo0.67 phases. But, for the Ti35Mo powder, it is not indicated the type of the crystalline structure corresponding to the single-phase Ti0.67Mo0.33. Please provide it.

Also, it should be developed a discussion about the reason of the variation of the presented phases in the XRD spectra, from Ti0.9Mo0.1 to Ti0.67Mo0.33. Few words about phase diagram, possible phases, solubility limits etc.

[8] - p.5, lines 205-209: this paragraph, alongside with Figure 2, indicates the decrease of the powder particle size in function of Mo content increase. Please indicate a reason for this decrease, even a probable one. Otherwise, it is a simple result presentation, without any comment/analysis/consequences/discussion.

[9] - in order to have a correlation between the text and the images, please indicate with distinct arrows on figure 8 the localization of the following structural constituents nominated in the text: polygonal grains of Ti (β) phase, the aggregated YSZ grains with spherical morphology, the areas of the grain boundaries of the mechanically alloyed particle of the matrix. The same for figure 9.

[10] - p.12, lines 260-261: “The presence of Fe atoms confirms the MA process impurity, however, Fe is also a Ti(β) stabilizing element and can be found in β-type alloys [41]. In this case, Fe is dissolved in Ti-Mo matrix and is not revealed on XRD spectra.” These phrases that refers to the presence/detection of iron on EDS analysis should not be superficially treated. From fig. 10 it results a presence of about 1% wt. Fe, so it cannot be negligible, even if is assigned as impurity. Secondly, even if iron is dissolved in the Ti-Mo matrix, it can influence the mechanical and structural properties of the solid solution not only as beta stabilizing element. The fact that it is not revealed on XRD spectra, just for being dissolved in solid solution, it’s not a reason to ignore or minimize the iron presence. For the present experimental work, the apparition of the iron in the final chemical composition is not accidental, considering the processing procedures (mechanical alloying/milling etc.). Therefore, in my opinion, there are two possibilities: either the iron presence and influence is analyzed in more detail, or a development of the subject is specified for future research, postponed for the time being. But not ignored.

[11] - p.12, lines 252-257: All written sentences should include specific references.

[12] - in my opinion, the part #3.1 it’s additional to the proposed objective, that is the study of the hall composite structure and not component powder individually. But, the main part, the # 3.2, that refers to the composite structure, is very superficial treated. All XRD spectra, figures 5-7, are presented (and not discussed) in three lines, p.8, lines 229-231. Why alpha phase is increasing? What about the evolution of the beta phase? What about other phases? Please provide a more complex and detail discussion.

[13] - p. 13, # 3.3: “The Young modulus and hardness presented a similar tendency.” Please revise the comments of this part, because the modulus and the hardness have not similar evolution: hardness values increase constantly with reinforcement phase increasing, instead of modulus that has the higher values for the 27-10 variant and not for “35” samples. Please provide also a possible explanation of these results.

[14] - p. 14, lines 290-294: Please provide specific references for the written sentences.

[15] - In conclusion, from all the experimental variants presented, there is no final global comparative result that can include all the studied properties and not separately, to offer (maybe) one or some perspective variants that deserve to be developed in the future or analyzed more carefully for other related properties concerning to the biocompatibility of the studied composites.

Author Response

Dear Reviewer 2,

We would like to appreciate you allowing us to resubmit our manuscript: The ultrafine-grain YSZ reinforced beta-titanium matrix composites to Metals. We are pleased with your comments and objective feedback considering the draft of our article. The changes were marked in our article in red colors.

All of the suggestions were included during the correction of our article and the comments were appropriately responded:

[1] - p.2, line84: 2-9% of Y2O3 instead of 2-9% of Y2O5

Thank you. It was corrected.

[2] - p.2, line 86: For being in conformity with Table 1, the phrase “based on titanium beta matrix with YSZ reinforcement” should be corrected in “based on titanium beta matrix with [YSZ and Y2O3] reinforcement” because, beside YSZ with variable content, it has been used also 1% of Y2O3 as reinforcement phase. That is understandable from Table 1. If no, please revise.

The authors agree with that comment. We have replaced “YSZ” phrase to “YSZ + Y2O3” phrase.

[3] - for the presented experiments, the introduction of a variable amount of YSZ in the composite approach is explained at the end of the introduction. However, it is not explained the additional presence of 1% of Y2O3, beside the YSZ. What was the reason?

We agree with the comment, additional information about the presence of Y2O3 was needed. Comments about it were added (see p.3 l.94-98). The authors wish also to explain that during preliminary research the investigation proceeds by bellow path:

  1. mechanical alloying of the matrix elements with reinforcement phase-phase disintegration after MA, after sintering no YSZ peaks in the structure
  2. mechanical alloying of the matrix elements and next mixing in a mortar with reinforcement phase until powder homogeneity- after sintering no YSZ peaks in the structure, phase dissolution during sintering
  3. mechanical alloying of the matrix elements and next mixing in a mortar with reinforcement phase until powder homogeneity, different sintering temperature and time settings – time and temperature variation don’t restrict YSZ phase dissolution during sintering additionally for shorter times and lower temperatures result in sinters porosity
  4. mechanical alloying of the matrix elements and next mixing in a mortar with reinforcement phase and 1% Y2O3 addition until powder homogeneity- after sintering YSZ peaks present in all examined examples however with the visible partial dissolution of YSZ

Additionally presented data that compare steps 2 and 4 where yttria oxide is added and no other processing parameters is changed.

The additional step of yttria oxide addition remains the author's original concept that as we can see from the collated data, shows expected improvement. Compared data differ from each other firstly by the YSZ peaks presence, Ti(α) phase peaks intensity ratio as also transitional cubic molybdenum based phases occurrence.

[4] - p.2, lines 63-64: “The disadvantage of titanium alloys is their low wear resistance [4]. It can be improved by creating composites based on titanium alloys.” It follows from this sentence that the wear resistance represents the mechanical property considered for the present research for which composite structures are created. Therefore, it must be explained why corrosion resistance and wettability were analyzed instead of wear resistance. The declaration from the final sentence of the introduction part “The influence of variable among of YSZ (3%, 5% and 10%) in composites on properties was investigated” is too general. What properties and for what reason? What was the goal for choosing specific properties to analyze? Please provide a detail explanation.

Thank you for this comment the introduction section was rewritten (see p.3 l.85-106)

[5] - in my opinion, the introduction part fails to present the specific results regarding the already use of YSZ in composite structures. What are the actual achieving and what can be done next?

Thank you for this comment the introduction section was rewritten (see p.3 l.99-106)

 [6] - The caption from figure 3 is not quite understandable. Please revise the expression in English.

Thank you for this comment the caption was rewritten.

[7] - p.5, lines 201-204: For the powders Ti23Mo and Ti27Mo, the authors indicate the cubic structure for the Ti0.9Mo0.1 and Ti0.33Mo0.67 phases. But, for the Ti35Mo powder, it is not indicated the type of the crystalline structure corresponding to the single-phase Ti0.67Mo0.33. Please provide it.

Thank you for your suggestion. We have added the information about the cubic structure of Ti0.67Mo0.33 phase.

Also, it should be developed a discussion about the reason of the variation of the presented phases in the XRD spectra, from Ti0.9Mo0.1 to Ti0.67Mo0.33. Few words about phase diagram, possible phases, solubility limits etc.

Thank you for this comment, a proper discussion and reference data were added.(see p.8 l. 242-257)

[8] - p.5, lines 205-209: this paragraph, alongside with Figure 2, indicates the decrease of the powder particle size in function of Mo content increase. Please indicate a reason for this decrease, even a probable one. Otherwise, it is a simple result presentation, without any comment/analysis/consequences/discussion.

Thank you for this comment, a proper discussion, and reference data were added.(see p.6 l.222-224)

[9] - in order to have a correlation between the text and the images, please indicate with distinct arrows on figure 8 the localization of the following structural constituents nominated in the text: polygonal grains of Ti (β) phase, the aggregated YSZ grains with spherical morphology, the areas of the grain boundaries of the mechanically alloyed particle of the matrix. The same for figure 9.

Thank you for this comment, a proper indication was placed in Fig. 8 and 9 the discussion, and Fig 8 caption was partly changed. (see p.12 l. 267-268 p.12 l.272-274)

 [10] - p.12, lines 260-261: “The presence of Fe atoms confirms the MA process impurity, however, Fe is also a Ti(β) stabilizing element and can be found in β-type alloys [41]. In this case, Fe is dissolved in Ti-Mo matrix and is not revealed on XRD spectra.” These phrases that refers to the presence/detection of iron on EDS analysis should not be superficially treated. From fig. 10 it results a presence of about 1% wt. Fe, so it cannot be negligible, even if is assigned as impurity. Secondly, even if iron is dissolved in the Ti-Mo matrix, it can influence the mechanical and structural properties of the solid solution not only as beta stabilizing element. The fact that it is not revealed on XRD spectra, just for being dissolved in solid solution, it’s not a reason to ignore or minimize the iron presence. For the present experimental work, the apparition of the iron in the final chemical composition is not accidental, considering the processing procedures (mechanical alloying/milling etc.). Therefore, in my opinion, there are two possibilities: either the iron presence and influence is analyzed in more detail, or a development of the subject is specified for future research, postponed for the time being. But not ignored.

Thank you for this comment, possible further plan of research was determined in the manuscript text (see p.13 286-289)

 [11] - p.12, lines 252-257: All written sentences should include specific references.

We appreciate that comment, the text was changed (see p.13 l.277-278)

[12] - in my opinion, the part #3.1 it’s additional to the proposed objective, that is the study of the hall composite structure and not component powder individually. But, the main part, the # 3.2, that refers to the composite structure, is very superficial treated. All XRD spectra, figures 5-7, are presented (and not discussed) in three lines, p.8, lines 229-231. Why alpha phase is increasing? What about the evolution of the beta phase? What about other phases? Please provide a more complex and detail discussion.

Thank you for this comment, a proper discussion, and reference data were added to the main manuscript text.(see p.8 l. 242-257)

[13] - p. 13, # 3.3: “The Young modulus and hardness presented a similar tendency.” Please revise the comments of this part, because the modulus and the hardness have not similar evolution: hardness values increase constantly with reinforcement phase increasing, instead of modulus that has the higher values for the 27-10 variant and not for “35” samples. Please provide also a possible explanation of these results.

Thank you for this comment a discussion was changed.(see p14 .l. 301-318)

[14] - p. 14, lines 290-294: Please provide specific references for the written sentences.

We appreciate that comment. We have added references to the main text.

[15] - In conclusion, from all the experimental variants presented, there is no final global comparative result that can include all the studied properties and not separately, to offer (maybe) one or some perspective variants that deserve to be developed in the future or analyzed more carefully for other related properties concerning to the biocompatibility of the studied composites

We appreciate that comment. The authors improve in general the main text (discussions, reference and corrections) for higher clearance of the drawn conclusions, some of the further steps for research were also pointed in the previous subsections.

We would like to thank you once again for your suggestions for improving our manuscript.

Yours faithfully,

Miklaszewski

Reviewer 3 Report

The author presents an interesting paper about the influence of YSZ addition on the mechanical and corrosion performance of Ti-Mo alloys. The topic has been addressed correctly but some important aspects need to be amended, modified, or incorporated.

I encourage the author to read carefully the following point to improve further submissions.

  1. There are 5 self-references from Miklaszewski. Please remove 2-3 of them.
  2. Introduction: As a suggestion, you might want to include the new HEA Ti-alloys which are now a popular topic (line 46), that could increase the visibility of this research work. Here are 2 references for your consideration

Materials 2020, 13(13), 3001; https://doi.org/10.3390/ma13133001 - 06 Jul 2020

Metals 2020, 10(11), 1463; https://doi.org/10.3390/met10111463 - 01 Nov 2020

  1. Introduction: Remove the term "in-depth" and replaced with "detailed".
  2. Materials and Methods: The author must provide detailed information regarding the mixing method (line 104).
  3. Please replaces the listing with a consecutive line (line 117)
  4. Change “by the” for “with” (line 138)
  5. Please replaces the listing with a consecutive line (line 147)
  6. Please replaces the listing with a consecutive line (line 182)
  7. Results and discussion: Change the term methods by routes (line 197)
  8. Why is the reason for the significant fragmentation of the powder? (line 209). Please include a suitable explanation.
  9. How did the author determine the increases in Ti (alfa) contend? (line231). Please add a few lines to explain this assertion.
  10. Replace “satisfying” with “effective” (line 247)
  11. Please include information on where exactly was taken micrograph of figure 9. Also, the average size of the aggregates could be present as 700 +/- 120. (line249).
  12. Authors report a value of 1 %wt of Fe for specimen 27-10 (line 260). The Fe content was about the same on all specimen analyzed? Please include such information.
  13. Please add some information regarding where the micrograph of figure 10 was taken. Also, the figure as outside the right margin, it must be fixed.
  14. The author must include a suitable explanation for the behavior exposed in lines 276-277. Also, what could be the effect of the grain size? It is very different among the specimens.
  15. Figure 11 was not explained sufficiently in the text, either include an extensive description or remove it.

Author Response

Dear Reviewer 3,

We would like to appreciate you allowing us to resubmit our manuscript: The ultrafine-grain YSZ reinforced beta-titanium matrix composites to Metals. We are pleased with your comments and objective feedback considering the draft of our article.

All of the suggestions were included during the correction of our article and the comments were appropriately responded:

There are 5 self-references from Miklaszewski. Please remove 2-3 of them.

Thank you for your suggestion. We have removed 2 references.

Introduction: As a suggestion, you might want to include the new HEA Ti-alloys which are now a popular topic (line 46), that could increase the visibility of this research work. Here are 2 references for your consideration

We agree that additional references increase the value of this research work. The authors have added one of the proposed references. (see p. 2 l. 51) 

Introduction: Remove the term "in-depth" and replaced with "detailed".

The authors agree with that comment. We have replaced “in-depth” phrase to “detailed” phrase. (see p. 2 l. 53)

Materials and Methods: The author must provide detailed information regarding the mixing method (line 104).

Thank you for your suggestion. We have added information that the powders were mixed using a mortar. (see p. 3 l. 120)

Please replaces the listing with a consecutive line (line 117)

Thank you for this comment, however, the authors decided to keep the data in the form of a list, because this form increases the readability of the text.

Change “by the” for “with” (line 138)

Thank you for this comment. We have made a change. (see p. 4 l. 156)

Please replaces the listing with a consecutive line (line 147)

We appreciate that comment. We change the style of the presented data. (see p. 5 l. 165-166)

Please replaces the listing with a consecutive line (line 182)

Thank you for your suggestion. We change the style of the presented data. (see p. 5 l. 198-200)

Results and discussion: Change the term methods by routes (line 197)

Thank you for this comment. We also thought about “technique” but at the end, we decided to stay with the term “methods”, because it is more comprehensible.

Why is the reason for the significant fragmentation of the powder? (line 209). Please include a suitable explanation.

Thank you for this comment. We have added a proper explanation (see p. 6 l. 222-224)

How did the author determine the increases in Ti (alfa) contend? (line231). Please add a few lines to explain this assertion.

Thank you for this comment. We added an explanation of this dependence (see p. 8 l. 245-250)

Replace “satisfying” with “effective” (line 247)

Thank you. It was corrected. (see p. 12 l. 272)

Please include information on where exactly was taken micrograph of figure 9. Also, the average size of the aggregates could be present as 700 +/- 120. (line249).

Thank you for your suggestion. We have added the information about the phases in Figure 9 and, also, we have changed the average size od aggregates. (see p. 12 l. 274)

Authors report a value of 1 %wt of Fe for specimen 27-10 (line 260). The Fe content was about the same on all specimen analyzed? Please include such information.

We appreciate that comment. We have added the information that the value of Fe content was on the same level in all cases of the matrix. (see p. 13 l. 286-289)

Please add some information regarding where the micrograph of figure 10 was taken. Also, the figure as outside the right margin, it must be fixed.

Thank you for your suggestion. We have added the information about the phases in Figure 10.

The author must include a suitable explanation for the behavior exposed in lines 276-277. Also, what could be the effect of the grain size? It is very different among the specimens.

Thank you for your suggestion a disusion was changed (see p. 14 l. 310-314)

Figure 11 was not explained sufficiently in the text, either include an extensive description or remove it.

Thank you for your suggestion. We decided to remove figure 11.

We would like to thank you once again for your suggestions for improving our manuscript.

Yours faithfully,

Miklaszewski

Round 2

Reviewer 2 Report

Dear Authors,

Thank you for considering all my comments and suggestions for your manuscript to be scientifically valid.

I appreciate and agree with all the corrections / modifications / additions made which, in their current form, provide a more valuable image of the entire manuscript.

Author Response

Thank you for your replay 

Reviewer 3 Report

Congratulations on your revision.

All the questions and remarks rise during reviewing were resolved in good shape.

There are still some minor suggestions for your consideration (below) and, once you've considered these, I look forward to accepting your paper for publication.

  • Check the figure alignment, size, and disposition.  
  • Run a deep spell check to resolve any mistake.

Author Response

Thank you for your reply. We check the text to find the spell issues and made some corrections also with the margins and figure alignment. The description on Fig 10 was corrected. We also wish to explain that during submission processing the original version of the manuscript was edited by the journal editors, for the view we attache the original previous version in PDF format.

We hope that the final version of the manuscript which includes your kind minor suggestion will now allow accepting our paper for publication in Metals.
